# Are Anchor Points Really Indispensable
# in Label-Noise Learning?

**Xiaobo Xia**[1,2]    **Tongliang Liu**[1]    **Nannan Wang**[2]
**Bo Han**[3]    **Chen Gong**[4]    **Gang Niu**[3]    **Masashi Sugiyama**[3,5]

[1]University of Sydney    [2]Xidian University    [3]RIKEN
[4]Nanjing University of Science and Technology    [5]University of Tokyo

## Abstract

In label-noise learning, the *noise transition matrix*, denoting the probabilities that clean labels flip into noisy labels, plays a central role in building *statistically consistent classifiers*. Existing theories have shown that the transition matrix can be learned by exploiting *anchor points* (i.e., data points that belong to a specific class almost surely). However, when there are no anchor points, the transition matrix will be poorly learned, and those previously consistent classifiers will significantly degenerate. In this paper, without employing anchor points, we propose a *transition-revision* ($T$-Revision) method to effectively learn transition matrices, leading to better classifiers. Specifically, to learn a transition matrix, we first initialize it by exploiting data points that are similar to anchor points, having high *noisy class posterior probabilities*. Then, we modify the initialized matrix by adding a *slack variable*, which can be learned and validated together with the classifier by using noisy data. Empirical results on benchmark-simulated and real-world label-noise datasets demonstrate that without using exact anchor points, the proposed method is superior to state-of-the-art label-noise learning methods.

## 1   Introduction

Label-noise learning can be dated back to [1] but becomes a more and more important topic recently. The reason is that, in this era, datasets are becoming bigger and bigger. Often, large-scale datasets are infeasible to be annotated accurately due to the expensive cost, which naturally brings us cheap datasets with noisy labels.

Existing methods for label-noise learning can be generally divided into two categories: algorithms that result in *statistically inconsistent/consistent* classifiers. Methods in the first category usually employ heuristics to reduce the side-effect of noisy labels. For example, many state-of-the-art approaches in this category are specifically designed to, e.g., select reliable examples [45, 14, 24], reweight examples [33, 15], correct labels [23, 17, 37, 32], employ side information [39, 21], and (implicitly) add regularization [13, 12, 43, 39, 21]. All those methods were reported to work empirically very well. However, the differences between the learned classifiers and the optimal ones for clean data are not guaranteed to vanish, i.e., no statistical consistency has been guaranteed.

The above issue motivates researchers to explore algorithms in the second category: *risk-/classifier-consistent* algorithms. In general, risk-consistent methods possess statistically consistent estimators to the clean risk (i.e., risk w.r.t. the clean data), while classifier-consistent methods guarantee the classifier learned from the noisy data is consistent to the optimal classifier (i.e., the minimizer of the clean risk) [42]. Methods in this category utilize the *noise transition matrix*, denoting the probabilities that clean labels flip into noisy labels, to build consistent algorithms. Let $Y$ denote the variable for the clean label, $\bar{Y}$ the noisy label, and $X$ the instance/feature. The basic idea is that given the

*noisy class posterior probability* $P(\bar{\mathbf{Y}}|X=x) = [P(\bar{Y}=1|X=x),\dots,P(\bar{Y}=C|X=x)]^\top$ (which can be learned using noisy data) and the transition matrix $T(X=x)$ where $T_{ij}(X=x) = P(\bar{Y}=j|Y=i,X=x)$, the *clean class posterior probability* $P(\mathbf{Y}|X=x)$ can be inferred, i.e., $P(\mathbf{Y}|X=x) = (T(X=x)^\top)^{-1}P(\bar{\mathbf{Y}}|X=x)$. For example, loss functions are modified to ensure risk consistency, e.g., [49, 17, 22, 29, 35, 26]; a noise adaptation layer is added to deep neural networks to design classifier-consistent deep learning algorithms [9, 30, 38, 47]. Those algorithms are strongly theoretically grounded but heavily rely on the success of learning transition matrices.

Given risk-consistent estimators, one stream to learn the transition matrix is the *cross-validation* method (using only noisy data) for binary classification [26]. However, it is prohibited for multi-class problems as its computational complexity grows exponentially to the number of classes. Besides, the current risk-consistent estimators involve the inverse of the transition matrix, making tuning the transition matrix inefficient and also leading to performance degeneration [30], especially when the transition matrix is non-invertible. Independent of risk-consistent estimators, another stream to learn the transition matrix is closely related to *mixture proportion estimation* [40]. A series of assumptions [36, 22, 35, 31] were proposed to efficiently learn transition matrices (or mixture parameters) by only exploiting the noisy data. All those assumptions require anchor points, i.e., instances belonging to a specific class with probability exactly one or close to one. Nonetheless, without anchor points, the transition matrix could be poorly learned, which will degenerate the accuracies of existing consistent algorithms.

Therefore, in this paper, to handle the applications where the anchor-point assumptions are violated [46, 41], we propose a *transition-revision* ($T$-Revision) method to effectively learn transition matrices, leading to better classifiers. In a high level, we design a deep-learning-based risk-consistent estimator to tune the transition matrix accurately. Specifically, we first initialize the transition matrix by exploiting examples that are similar to anchor points, namely, those having high estimated *noisy class posterior probabilities*. Then, we modify the initial matrix by adding a *slack variable*, which will be learned and validated together with the classifier by using noisy data only. Note that given true transition matrix, the proposed estimator will converge to the classification risk w.r.t. clean data by increasing the size of noisy training examples. Our heuristic for tuning the transition matrix is that a favorable transition matrix would make the classification risk w.r.t. clean data small. We empirically show that the proposed $T$-Revision method will enable tuned transition matrices to be closer to the ground truths, which explains why $T$-Revision is much superior to state-of-the-art algorithms in classification.

The rest of the paper is organized as follows. In Section 2 we review label-noise learning with anchor points. In Section 3, we discuss how to learn the transition matrix and classifier without anchor points. Experimental results are provided in Section 4. Finally, we conclude the paper in Section 5.

## 2  Label-Noise Learning with Anchor Points

In this section, we briefly review label-noise learning when there are anchor points.

**Preliminaries** Let $D$ be the distribution of a pair of random variables $(X, Y) \in \mathcal{X} \times \{1, 2, \dots, C\}$, where the feature space $\mathcal{X} \subseteq \mathbb{R}^d$ and $C$ is the size of label classes. Our goal is to predict a label $y$ for any given instance $x \in \mathcal{X}$. However, in many real-world classification problems, training examples drawn independently from distribution $D$ are unavailable. Before being observed, their true labels are independently flipped and what we can obtain is a noisy training sample $\{(X_i, \bar{Y}_i)\}_{i=1}^n$, where $\bar{Y}$ denotes the noisy label. Let $\bar{D}$ be the distribution of the noisy random variables $(X, \bar{Y}) \in \mathcal{X} \times \{1, 2, \dots, C\}$.

**Transition matrix** The random variables $\bar{Y}$ and $Y$ are related through a *noise transition matrix* $T \in [0,1]^{C \times C}$ [8]. Generally, the transition matrix depends on instances, i.e., $T_{ij}(X=x) = P(\bar{Y}=j|Y=i,X=x)$. Given only noisy examples, the *instance-dependent* transition matrix is *non-identifiable* without any additional assumption. For example, $P(\bar{Y}=j|X=x) = \sum_{i=1}^C T_{ij}(X=x)P(Y=i|X=x) = \sum_{i=1}^C T'_{ij}(X=x)P'(Y=i|X=x)$ are both valid, when $T'_{ij}(X=x) = T_{ij}(X=x)P(Y=i|X=x)/P'(\bar{Y}=i|X=x)$. In this paper, we study the *class-dependent* and *instance-independent* transition matrix, i.e., $P(\bar{Y}=j|Y=i,X=x) = P(\bar{Y}=j|Y=i)$, which is identifiable under mild conditions and on which the vast majority of current methods focus [14, 13, 30, 29, 26].

**Consistent algorithms** The transition matrix bridges the class posterior probabilities for noisy and clean data, i.e., $P(\bar{Y} = j|X = x) = \sum_{i=1}^{C} T_{ij} P(Y = i|X = x)$. Thus, it has been exploited to build consistent algorithms. Specifically, it has been used to modify loss functions to build *risk-consistent* estimators, e.g., [26, 35, 30], and has been used to correct hypotheses to build *classifier-consistent* algorithms, e.g., [9, 30, 47]. Note that an estimator is risk-consistent if, by increasing the size of noisy examples, the *empirical risk* calculated by noisy examples and the modified loss function will converge to the *expected risk* calculated by clean examples and the original loss function. Similarly, an algorithm is classifier-consistent if, by increasing the size of noisy examples, the learned classifier will converge to the optimal classifier learned by clean examples. Definitions of the expected and empirical risks can be found in Appendix B, where we further discuss how consistent algorithms work.

**Anchor points** The successes of consistent algorithms rely on firm bridges, i.e., accurately learned transition matrices. To learn transition matrices, the concept of anchor point was proposed [22, 35]. Anchor points are defined in the clean data domain, i.e., an instance $x$ is an anchor point for the class $i$ if $P(Y = i|X = x)$ is equal to one or close to one[1]. Given an $x$, if $P(Y = i|X = x) = 1$, we have that for $k \neq i, P(Y = k|X = x) = 0$. Then, we have

$$P(\bar{Y} = j|X = x) = \sum_{k=1}^{C} T_{kj} P(Y = k|X = x) = T_{ij}. \tag{1}$$

Namely, $T$ can be obtained via estimating the noisy class posterior probabilities for anchor points [47]. However, the requirement of given anchor points is a bit strong. Thus, anchor points are assumed to exist but unknown in datasets, which can be identified either theoretically [22] or heuristically [30].

Transition matrix learning is also closely related to *mixture proportion estimation* [40], which is independent of classification. By giving only noisy data, to ensure the *learnability* and efficiency of learning transition matrices (or mixture parameters), a series of assumptions were proposed, e.g., *irreducibility* [36], *anchor point* [22, 35], and *separability* [31]. All those assumptions require anchor points or instances belonging to a specific class with probability one or approaching one.

When there are no anchor points in datasets/data distributions, all the above mentioned methods will lead to inaccurate transition matrices, which will degenerate the performances of current consistent algorithms. This motivates us to investigate how to maintain the efficacy of those consistent algorithms without using exact anchor points.

## 3 Label-Noise Learning without Anchor Points

This section presents a deep-learning-based risk-consistent estimator for the classification risk w.r.t. clean data. We employ this estimator to tune the transition matrix effectively without using anchor points, which finally leads to better classifiers.

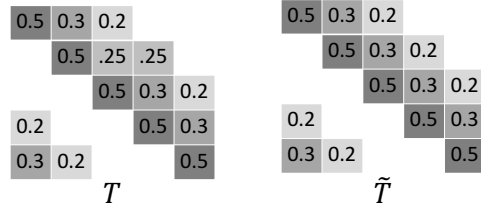

Figure 1: Illustrative experimental results (using a 5-class classification problem as an example). The noisy class posterior probability $P(\bar{\mathbf{Y}}|X = x)$ can be estimated by exploiting noisy data. Let an example have $P(\bar{\mathbf{Y}}|X = x) = [0.141; 0.189; 0.239; 0.281; 0.15]$. If the true transition matrix $T$ is given, we can infer the clean class posterior probability as $P(\mathbf{Y}|X = x) = (T^\top)^{-1} P(\bar{\mathbf{Y}}|X = x) = [0.15; 0.28; 0.25; 0.3; 0.02]$ and that the instance belongs to the fourth class. However, if the transition matrix is not accurately learned as $\tilde{T}$ (only slightly differing from $T$ with two entries in the second row), the clean class posterior probability can be inferred as $P(\mathbf{Y}|X = x) = (\tilde{T}^\top)^{-1} P(\bar{\mathbf{Y}}|X = x) = [0.1587; 0.2697; 0.2796; 0.2593; 0.0325]$ and the instance could be mistakenly classified into the third class.

## 3.1 Motivation

According to Eq. (1), to learn the transition matrix, $P(\bar{\mathbf{Y}}|X = x)$ needs to be estimated and anchor points need to be given. Note that learning $P(\bar{\mathbf{Y}}|X = x)$ may introduce error. Even worse, when there are no anchor points, it will be problematic if we use existing methods [36, 22, 35, 31] to learn transition matrices. For example, let $P(\mathbf{Y}|X = x^i)$ be the $i$-th column of a matrix $L$, $i = 1, \ldots, C$. If $x^i$ is an anchor point for the $i$-th class, then $L$ is an identity matrix. According to Eq. (1), if we use $x^i$ as an anchor point for the $i$-th class while $P(Y = i|X = x^i) \neq 1$ (e.g., the identified instances in [30] are not guaranteed to be anchor points), the learned transition matrix would be $TL$, where $L$ is a non-identity matrix. This means that transition matrices will be inaccurately estimated.

Based on inaccurate transition matrices, the accuracy of current consistent algorithms will significantly degenerate. To demonstrate this, Figure 1 shows that given a noisy class posterior probability $P(\bar{\mathbf{Y}}|X = x)$, even if the transition matrix changes slightly by two entries, e.g., $\|T - \tilde{T}\|_1/\|T\|_1 = 0.02$ where $T$ and $\tilde{T}$ are defined in Figure 1 and $\|T\|_1 = \sum_{ij} |T_{ij}|$, the inferred class posterior probability for the clean data may lead to an incorrect classification. Since anchor points require clean class posterior probabilities to be or approach one, which is quite strong to some real-world applications [46, 41], we would like to study how to maintain the performances of current consistent algorithms when there are no anchor points and then transition matrices are inaccurately learned.

## 3.2 Risk-consistent estimator

Intuitively, the entries of transition matrix can be tuned by minimizing the risk-consistent estimator, since the estimator is asymptotically identical to the expected risk for the clean data and that a favorable transition matrix should make the clean expected risk small. However, existing risk-consistent estimators involve the inverse of transition matrix (more details are provided in Appendix B), which degenerates classification performances [30] and makes tuning the transition matrix ineffectively. To address this, we propose a risk-consistent estimator that does not involve the inverse of the transition matrix.

The inverse of transition matrix is involved in risk-consistent estimators, since the noisy class posterior probability $P(\bar{\mathbf{Y}}|X = x)$ and the transition matrix are explicitly or implicitly used to infer the clean class posterior probability $P(\mathbf{Y}|X = x)$, i.e., $P(\mathbf{Y}|X = x) = (T^\top)^{-1}P(\bar{\mathbf{Y}}|X = x)$. To avoid the inverse in building risk-consistent estimators, we directly estimate $P(\mathbf{Y}|X = x)$ instead of inferring it through $P(\bar{\mathbf{Y}}|X = x)$. Thanks to the equation $T^\top P(\mathbf{Y}|X = x) = P(\bar{\mathbf{Y}}|X = x)$, $P(\mathbf{Y}|X = x)$ and $P(\bar{\mathbf{Y}}|X = x)$ could be estimated at the same time by adding the true transition matrix to modify the output of the softmax function, e.g., [47, 30]. Specifically, $P(\bar{\mathbf{Y}}|X = x)$ can be learned by exploiting the noisy data, as shown in Figure 2 by minimizing the unweighted loss $\bar{R}_n(f) = 1/n \sum_{i=1}^n \ell(f(X_i), \bar{Y}_i)$, where $\ell(f(X), \bar{Y})$ is a *loss function* [25]. Let $\hat{T} + \Delta T$ be the true transition matrix, i.e., $\hat{T} + \Delta T = T$. Due to $P(\bar{\mathbf{Y}}|X = x) = T^\top P(\mathbf{Y}|X = x)$, the output of the softmax function $g(x) = \hat{P}(\mathbf{Y}|X = x)$ before the transition matrix is an approximation for $P(\mathbf{Y}|X = x)$. However, the learned $g(x) = \hat{P}(\mathbf{Y}|X = x)$ by minimizing the unweighted loss may perform poorly if the true transition matrix is inaccurately learned as explained in the motivation.

If having $P(\mathbf{Y}|X = x)$ and $P(\bar{\mathbf{Y}}|X = x)$, we could employ the *importance reweighting* technique [11, 22] to rewrite the expected risk w.r.t. clean data without involving the inverse of transition matrix. Specifically,

$$R(f) = \mathbb{E}_{(X,Y)\sim D}[\ell(f(X), Y)] = \int_x \sum_i P_D(X = x, Y = i)\ell(f(x), i)dx$$

$$= \int_x \sum_i P_{\bar{D}}(X = x, \bar{Y} = i)\frac{P_D(X = x, \bar{Y} = i)}{P_{\bar{D}}(X = x, \bar{Y} = i)}\ell(f(x), i)dx$$

$$= \int_x \sum_i P_{\bar{D}}(X = x, \bar{Y} = i)\frac{P_D(\bar{Y} = i|X = x)}{P_{\bar{D}}(\bar{Y} = i|X = x)}\ell(f(x), i)dx \tag{2}$$

$$= \mathbb{E}_{(X,Y)\sim \bar{D}}[\bar{\ell}(f(X), Y)],$$

where $D$ denotes the distribution for clean data, $\bar{D}$ for noisy data, $\bar{\ell}(f(x), i) = \frac{P_D(\bar{Y}=i|X=x)}{P_{\bar{D}}(\bar{Y}=i|X=x)}\ell(f(x), i)$, and the second last equation holds because label noise is assumed to be

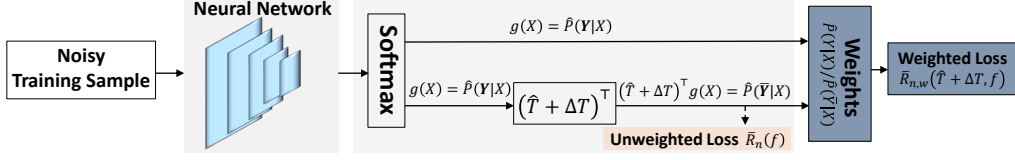

Figure 2: An overview of the proposed method. The proposed method will learn a more accurate classifier because the transition matrix is renovated.

---

**Algorithm 1** Reweight $T$-Revision (Reweight-R) Algorithm.

---
**Input**: Noisy training sample $\mathcal{D}_t$; Noisy validation set $\mathcal{D}_v$.
**Stage 1: Learn $\hat{T}$**
1: Minimize the unweighted loss to learn $\hat{P}(\bar{\mathbf{Y}}|X = x)$ without a noise adaption layer;
2: Initialize $\hat{T}$ according to Eq. (1) by using instances with the highest $\hat{P}(\bar{Y} = i|X = x)$ as anchor points for the $i$-th class;
**Stage 2: Learn the classifier $f$ and $\Delta T$**
3: Initialize the neural network by minimizing the weighted loss with a noisy adaption layer $\hat{T}^\top$;
4: Minimize the weighted loss to learn $f$ and $\Delta T$ with a noisy adaption layer $(\hat{T} + \Delta T)^\top$;
//Stopping criterion for learning $\hat{P}(\bar{\mathbf{Y}}|X = x)$, $f$ and $\Delta T$: when $\hat{P}(\bar{\mathbf{Y}}|X = x)$ yields the minimum classification error on the noisy validation set $\mathcal{D}_v$
**Output**: $\hat{T}$, $\Delta T$, and $f$.

---

independent of instances. In the rest of the paper, we have omitted the subscript for $P$ when no confusion is caused. Since $P(\bar{\mathbf{Y}}|X = x) = T^\top P(\mathbf{Y}|X = x)$ and that the diagonal entries of (learned) transition matrices for label-noise learning are all much larger than zero, $P_D(\bar{Y} = i|X = x) \neq 0$ implies $P_{\bar{D}}(\bar{Y} = i|X = x) \neq 0$, which also makes the proposed importance reweighting method stable without truncating the importance ratios.

Eq. (2) shows that the expected risk w.r.t. clean data and the loss $\ell(f(x), i)$ is equivalent to an expected risk w.r.t. noisy data and a reweighted loss, i.e., $\frac{P_D(\bar{Y}=i|X=x)}{P_{\bar{D}}(\bar{Y}=i|X=x)} \ell(f(x), i)$. The empirical counterpart of the risk in the rightmost-hand side of Eq. (2) is therefore a risk-consistent estimator for label-noise learning. We exploit a deep neural network to build this counterpart. As shown in Figure 2, we use the output of the softmax function $g(x)$ to approximate $P(\mathbf{Y}|X = x)$, i.e., $g(x) = \hat{P}(\mathbf{Y}|X = x) \approx P(\mathbf{Y}|X = x)$. Then, $T^\top g(x)$ (or $(\hat{T} + \Delta T)^\top g(x)$ in the figure) is an approximation for $P(\bar{\mathbf{Y}}|X = x)$, i.e., $T^\top g(x) = \hat{P}(\bar{\mathbf{Y}}|X = x) \approx P(\bar{\mathbf{Y}}|X = x)$. By employing $\hat{P}(Y = y|X = x)/\hat{P}(\bar{Y} = y|X = x)$ as weight, we build the risk-consistent estimator as

$$\bar{R}_{n,w}(T, f) = \frac{1}{n}\sum_{i=1}^{n} \frac{g_{\bar{Y}_i}(X_i)}{(T^\top g)_{\bar{Y}_i}(X_i)} \ell(f(X_i), \bar{Y}_i), \tag{3}$$

where $f(X) = \arg\max_{j \in \{1,\dots,C\}} g_j(X)$, $g_j(X)$ is an estimate for $P(Y = j|X)$, and the subscript $w$ denotes that the loss function is weighted. Note that if the true transition matrix $T$ is given, $\bar{R}_{n,w}(T, f)$ only has one argument $g$ to learn.

### 3.3 Implementation and the $T$-revision method

When the true transition matrix $T$ is unavailable, we propose to use $\bar{R}_{n,w}(\hat{T} + \Delta T, f)$ to approximate $R(f)$, as shown in Figure 2. To minimize $\bar{R}_{n,w}(\hat{T} + \Delta T, f)$, a two-stage training procedure is proposed. Stage 1: first learn $P(\bar{\mathbf{Y}}|X = x)$ by minimizing the unweighted loss without a noise adaption layer and initialize $\hat{T}$ by exploiting examples that have the highest learned $\hat{P}(\bar{\mathbf{Y}}|X = x)$; Stage 2: modify the initialization $\hat{T}$ by adding a slack variable $\Delta T$ and learn the classifier and $\Delta T$ by minimizing the weighted loss. The procedure is called the Weighted $T$-Revision method and is summarized in Algorithm 1. It is worthwhile to mention that all anchor points based consistent estimators for label-noise learning have a similar two-stage training procedure. Specifically, with one stage to learn $P(\bar{\mathbf{Y}}|X = x)$ and the transition matrix and a second stage to learn the classifier for the clean data.

The proposed $T$-revision method works because we learn $\Delta T$ by minimizing the risk-consistent estimator, which is asymptotically equal to the expected risk w.r.t. clean data. The learned slack variable can also be validated on the noisy validation set, i.e., to check if $\hat{P}(\bar{\mathbf{Y}}|X = x)$ fits the validation set. The philosophy of our approach is similar to that of the cross-validation method. However, the proposed method does not need to try different combinations of parameters ($\Delta T$ is learned) and thus is much more computationally efficient. Note that the proposed method will also boost the performances of consistent algorithms even there are anchor points as the transition matrices and classifiers are jointly learned. Note also that if a clean validation set is available, it can be used to better initialize the transition matrix, to better validate the slack variable $\Delta T$, and to fine-tune the deep network.

### 3.4 Generalization error

While we have discussed the use of the proposed estimator for evaluating the risk w.r.t clean data, we theoretically justify how it generalizes for learning classifiers. Assume the neural network has $d$ layers, parameter matrices $W_1, \ldots, W_d$, and activation functions $\sigma_1, \ldots, \sigma_{d-1}$ for each layer. Let denote the mapping of the neural network by $h : x \mapsto W_d \sigma_{d-1}(W_{d-1}\sigma_{d-2}(\ldots \sigma_1(W_1 x))) \in \mathbb{R}^C$. Then, the output of the softmax is defined by $g_i(x) = \exp(h_i(x))/\sum_{k=1}^{C} \exp(h_k(x)), i = 1, \ldots, C$. Let $\hat{f} = \arg\max_{i \in \{1,\ldots,C\}} \hat{g}_i$ be the classifier learned from the hypothesis space $F$ determined by the real-valued parameters of the neural network, i.e., $\hat{f} = \arg\min_{f \in F} \bar{R}_{n,w}(f)$.

To derive a generalization bound, as the common practice [6, 25], we assume that instances are upper bounded by $B$, i.e., $\|x\| \leq B$ for all $x \in \mathcal{X}$, and that the loss function is $L$-Lipschitz continuous w.r.t. $f(x)$ and upper bounded by $M$, i.e., for any $f_1, f_2 \in F$ and any $(x, \bar{y})$, $|\ell(f_1(x), \bar{y}) - \ell(f_2(x), \bar{y})| \leq L|f_1(x) - f_2(x)|$, and for any $(x, \bar{y})$, $\ell(f(x), \bar{y}) \leq M$.

**Theorem 1** *Assume the Frobenius norm of the weight matrices $W_1, \ldots, W_d$ are at most $M_1, \ldots, M_d$. Let the activation functions be 1-Lipschitz, positive-homogeneous, and applied element-wise (such as the ReLU). Let the loss function be the cross-entropy loss, i.e., $\ell(f(x), \bar{y}) = -\sum_{i=1}^{C} 1_{\{\bar{y}=i\}} \log(g_i(x))$. Let $\hat{f}$ and $\Delta\hat{T}$ be the learned classifier and slack variable. Assume $\Delta\hat{T}$ is searched from a space of $\Delta T$ constituting valid transition matrices[2], i.e., $\forall \Delta T$ and $\forall i \neq j$, $\hat{T}_{ij} + \Delta T_{ij} \geq 0$ and $\hat{T}_{ii} + \Delta T_{ii} > \hat{T}_{ij} + \Delta T_{ij}$. Then, for any $\delta > 0$, with probability at least $1 - \delta$,*

$$\mathbb{E}[\bar{R}_{n,w}(\hat{T} + \Delta\hat{T}, \hat{f})] - \bar{R}_{n,w}(\hat{T} + \Delta\hat{T}, \hat{f}) \leq \frac{2BCL(\sqrt{2d\log 2} + 1)\Pi_{i=1}^{d} M_i}{\sqrt{n}} + CM\sqrt{\frac{\log 1/\delta}{2n}}.$$

A detailed proof is provided in Appendix C. The factor $(\sqrt{2d\log 2} + 1)\Pi_{i=1}^{d} M_i$ is induced by the hypothesis complexity of the deep neural network [10] (see Theorem 1 therein), which could be improved [27, 48, 16]. Although the proposed reweighted loss is more complex than the traditional unweighted loss function, we have derived a generalization error bound not larger than those derived for the algorithms employing the traditional loss [25] (can be seen by Lemma 2 in the proof of the theorem). This shows that the proposed Algorithm 1 does not need a larger training sample to achieve a small difference between training error ($\bar{R}_{n,w}(\hat{T} + \Delta\hat{T}, \hat{f})$) and test error ($\mathbb{E}[\bar{R}_{n,w}(\hat{T} + \Delta\hat{T}, \hat{f})]$). Also note that deep learning is powerful in yielding a small training error. If the training sample size $n$ is large, then the upper bound in Theorem 1 is small, which implies a small $\mathbb{E}[\bar{R}_{n,w}(\hat{T} + \Delta\hat{T}, \hat{f})]$ and justifies why the proposed method will have small test errors in the experiment section. Meanwhile, in the experiment section, we show that the proposed method is much superior to the state-of-the-art methods in classification accuracy, implying that the small generalization error is not obtained at the cost of enlarging the approximation error.

## 4 Experiments

**Datasets** We verify the effectiveness of the proposed method on three synthetic noisy datasets, i.e., *MNIST* [19], *CIFAR-10* [18], and *CIFAR-100* [18], and one real-world noisy dataset, i.e., *clothing1M*

Table 1: Means and standard deviations (percentage) of classification accuracy. Methods with "-A" means that they run on the intact datasets without removing possible anchor points; Methods with "-R" means that the transition matrix used is revised by a revision $\Delta\hat{T}$.

| | MNIST | | CIFAR-10 | | CIFAR-100 | |
|---|---|---|---|---|---|---|
| | Sym-20% | Sym-50% | Sym-20% | Sym-50% | Sym-20% | Sym-50% |
| Decoupling-A | 95.39±0.29 | 81.52±0.29 | 79.85±0.30 | 52.22±0.45 | 42.75±0.49 | 29.24±0.54 |
| MentorNet-A | 96.57±0.18 | 90.13±0.09 | 80.49±0.52 | 70.71±0.24 | 52.11±0.10 | 38.45±0.25 |
| Co-teaching-A | 97.22±0.18 | 91.68±0.21 | 82.38±0.11 | 72.80±0.45 | 54.23±0.08 | 41.37±0.08 |
| Forward-A | 98.75±0.08 | 97.86±0.22 | 85.63±0.52 | 77.92±0.66 | 57.75±0.37 | 44.66±1.01 |
| Reweight-A | 98.71±0.11 | 98.13±0.19 | 86.77±0.40 | 80.16±0.46 | 58.35±0.64 | 43.97±0.67 |
| Forward-A-R | 98.84±0.09 | 98.12±0.22 | 88.10±0.21 | 81.11±0.74 | 62.13±2.09 | **50.46±0.52** |
| Reweight-A-R | **98.91±0.04** | **98.38±0.21** | **89.63±0.13** | **83.40±0.65** | **65.40±1.07** | 50.24±1.45 |

Table 2: Means and standard deviations (percentage) of classification accuracy. Methods with "-N/A" means instances with high estimated $P(Y|X)$ are removed from the dataset; Methods with "-R" means that the transition matrix used is revised by a revision $\Delta\hat{T}$.

| | MNIST | | CIFAR-10 | | CIFAR-100 | |
|---|---|---|---|---|---|---|
| | Sym-20% | Sym-50% | Sym-20% | Sym-50% | Sym-20% | Sym-50% |
| Decoupling-N/A | 95.93±0.21 | 82.55±0.39 | 75.37±1.24 | 47.19±0.19 | 39.59±0.42 | 24.04±1.19 |
| MentorNet-N/A | 97.11±0.09 | 91.44±0.25 | 78.51±0.31 | 67.37±0.30 | 48.62±0.43 | 33.53±0.31 |
| Co-teaching-N/A | 97.69±0.23 | 93.58±0.49 | 81.72±0.14 | 70.44±1.01 | 53.21±0.54 | 40.06±0.83 |
| Forward-N/A | 98.64±0.12 | 97.74±0.13 | 84.75±0.81 | 74.32±0.69 | 56.23±0.34 | 39.28±0.59 |
| Reweight-N/A | 98.69±0.08 | 98.05±0.22 | 85.53±0.26 | 77.70±1.00 | 56.60±0.71 | 39.28±0.71 |
| Forward-N/A-R | 98.80±0.06 | 97.96±0.13 | 86.93±0.39 | 77.14±0.65 | 58.72±0.45 | 44.60±0.79 |
| Reweight-N/A-R | **98.85±0.02** | **98.37±0.17** | **88.90±0.22** | **81.55±0.94** | **62.00±1.78** | **44.75±2.10** |

[44]. *MNIST* has 10 classes of images including 60,000 training images and 10,000 test images. *CIFAR-10* has 10 classes of images including 50,000 training images and 10,000 test images. *CIFAR-100* also has 50,000 training images and 10,000 test images, but 100 classes. For all the datasets, we leave out 10% of the training examples as a validation set. The three datasets contain clean data. We corrupted the training and validation sets manually according to true transition matrices $T$. Specifically, we employ the symmetry flipping setting defined in Appendix D. Sym-50 generates heavy label noise and leads almost half of the instances to have noisy labels, while Sym-20 generates light label noise and leads around 20% of instances to have label noise. Note that the pair flipping setting [14], where each row of the transition matrix only has two non-zero entries, has also been widely studied. However, for simplicity, we do not pose any constraint on the slack variable $\Delta T$ to achieve specific speculation of the transition matrix, e.g., sparsity [13]. We leave this for future work.

Besides reporting the classification accuracy on test set, we also report the discrepancy between the learned transition matrix $\hat{T} + \Delta\hat{T}$ and the true one $T$. All experiments are repeated five times on those three datasets. *Clothing1M* consists of 1M images with real-world noisy labels, and additional 50k, 14k, 10k images with clean labels for training, validation, and testing. We use the 50k clean data to help initialize the transition matrix as did in the baseline [30].

**Network structure and optimization** For fair comparison, we implement all methods with default parameters by PyTorch on NVIDIA Tesla V100. We use a LeNet-5 network for *MNIST*, a ResNet-18 network for *CIFAR-10*, a ResNet-34 network for *CIFAR-100*. For learning the transition matrix $\hat{T}$ in the first stage, we follow the optimization method in [30]. During the second stage, we first use SGD with momentum 0.9, weight decay $10^{-4}$, batch size 128, and an initial learning rate of $10^{-2}$ to initialize the network. The learning rate is divided by 10 after the 40th epoch and 80th epoch. 200 epochs are set in total. Then, the optimizer and learning rate are changed to Adam and $5 \times 10^{-7}$ to learn the classifier and slack variable. For *CIFAR-10* and *CIFAR-100*, we perform data augmentation by horizontal random flips and $32 \times 32$ random crops after padding 4 pixels on each side. For *clothing1M*, we use a ResNet-50 pre-trained on ImageNet. Follow [30], we also exploit the 1M noisy data and 50k clean data to initialize the transition matrix. In the second stage, for initialization, we use SGD with momentum 0.9, weight decay $10^{-3}$, batch size 32, and run with learning rates $10^{-3}$ and $10^{-4}$ for 5 epochs each. For learning the classifier and slack variable, Adam is used and the learning rate is changed to $5 \times 10^{-7}$.

Table 3: Means and standard deviations (percentage) of classification accuracy on *MNIST* with different label noise levels. Methods with "-A" means that they run on the intact datasets without removing possible anchor points; Methods with "-R" means that the transition matrix used is revised by a revision $\Delta\hat{T}$; Methods with "-N/A" means instances with high estimated $P(Y|X)$ are removed from the dataset.

|  | Sym-60% | Sym-70% | Sym-80% |
|---|---|---|---|
| Forward-A | 97.10±0.08 | 96.06±0.41 | 91.46±1.03 |
| Forward-A-R | 97.65±0.11 | 96.42±0.35 | 91.77±0.22 |
| Reweight-A | 97.39±0.27 | 96.25±0.26 | 93.79±0.52 |
| Reweight-A-R | **97.83±0.18** | **97.13±0.08** | **94.19±0.45** |
| Forward-N/A | 96.82±0.14 | 94.61±0.28 | 85.95±1.01 |
| Forward-N/A-R | 96.99±0.16 | 95.02±0.17 | 86.04±1.03 |
| Reweight-N/A | 97.01±0.20 | 95.94±0.14 | 91.59±0.70 |
| Reweight-N/A-R | **97.81±0.12** | **96.59±0.15** | **91.91±0.65** |

Table 4: Classification accuracy (percentage) on *Clothing1M*.

| Decoupling | MentorNet | Co-teaching | Forward | Reweight | Forward-R | Reweight-R |
|---|---|---|---|---|---|---|
| 53.98 | 56.77 | 58.68 | 71.79 | 70.95 | 72.25 | **74.18** |

**Baselines** We compare the proposed method with state-of-the-art approaches. Specifically, we compare with the following three inconsistent but well-designed algorithms: Decoupling [24], MentorNet [15], and Co-teaching [14], which free the learning of transition matrices. To compare with consistent estimators, we set Forward [30], a classifier-consistent algorithm, and the importance reweighting method (Reweight), a risk-consistent algorithm, as baselines. The risk-consistent estimator involving the inverse of transition matrix, e.g., Backward in [30], has not been included in the comparison, because it has been reported to perform worse than the Forward method [30].

## 4.1 Comparison for classification accuracy

**The importance of anchor points** To show the importance of anchor points, we modify the datasets by moving possible anchor points, i.e., instances with large estimated class posterior probability $P(Y|X)$, before corrupting the training and validation sets. As the *MNIST* dataset is simple, we removed 40% of the instances with the largest estimated class posterior probabilities in each class. For *CIFAR-10* and *CIFAR-100*, we removed 20% of the instances with the largest estimated class posterior probabilities in each class. To make it easy for distinguishing, we mark a "-A" in the algorithm's name if it runs on the original intact datasets, and mark a "-N/A" in the algorithm's name if it runs on those modified datasets.

Comparing Decoupling-A, MentorNet-A, and Co-teaching-A in Table 1 with Decoupling-N/A, MentorNet-N/A, and Co-teaching-N/A in Table 2, we can find that on *MNIST*, the methods with "-N/A" work better; while on *CIFAR-10* and *CIFAR-100*, the methods with "-A" work better. This is because those methods are independent of transition matrices but dependent of dataset properties. Removing possible anchors points may not always lead to performance degeneration.

Comparing Forward-A and Reweight-A with Forward-N/A and Reweight-N/A, we can find that the methods without anchor points, i.e., with "-N/A", degenerate clearly. The degeneration on *MNIST* is slight because the dataset can be well separated and many instances have high class posterior probability even in the modify dataset. Those results show that, without anchor points, the consistent algorithms will have performance degeneration. Specifically, on *CIFAR-100*, the methods with "-N/A" have much worse performance than the ones with "-A", with accuracy dropping at least 4%.

To discuss the model performances on *MNIST* with more label noise, we raise the noise rates to 60%, 70%, 80%. Other experiment settings are unchanged. The results are presented in Table 3. We can see that the proposed model outperforms the baselines more significantly as the noise rate grows.

**Risk-consistent estimator vs. classifier-consistent estimator** Comparing Forward-A with Reweight-A in Table 1 and comparing Forward-N/A with Reweight-N/A in Table 2, it can be seen that the proposed Reweight method, a risk-consistent estimator not involving the inverse of transition matrix, works slightly better than or is comparable to Forward, a classifier-consistent

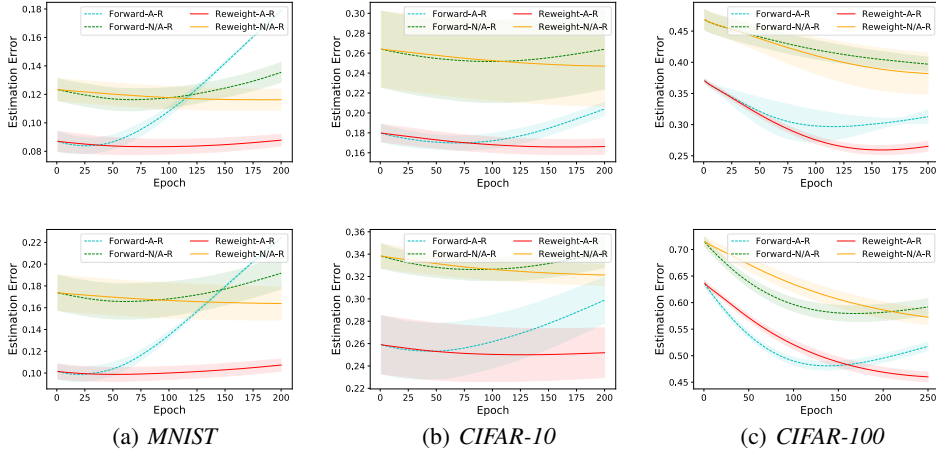

(a) *MNIST*    (b) *CIFAR-10*    (c) *CIFAR-100*

Figure 3: The estimation error of the transition matrix by employing classifier-consistent and risk-consistent estimators. The first row is about sym-20 label noise while the second row is about sym-50 label noise. The error bar for standard deviation in each figure has been shaded.

algorithm. Note that in [30], it is reported that Backward, a risk-consistent estimator which involves the inverse of the transition matrix, works worse than Forward, the classifier-consistent algorithm.

**The importance of $T$-revision** Note that for fair comparison, we also set it as a baseline to modify the transition matrix in Forward. As shown in Tables 1 and 2, methods with "-R" means that they use the proposed $T$-revision method, i.e., modify the learned $\hat{T}$ by adding $\Delta\hat{T}$. Comparing the results in Tables 1 and 2, we can find that the $T$-revision method significantly outperforms the others. Among them, the proposed Reweight-R works significantly better than the baseline Forward-R. We can find that the $T$-Revision method boosts the classification performance even without removing possible anchor points. The rationale behind this may be that the network, transition matrix, and classifier are jointly learned and validated and that the identified anchor points are not reliable.

**Comparison on real-world dataset** The proposed $T$-revision method significantly outperforms the baselines as shown in Table 4, where the highest accuracy is bold faced.

## 4.2 Comparison for estimating transition matrices

To show that the proposed risk-consistent estimator is more effective in modifying the transition matrix, we plot the estimation error for the transition matrix, i.e., $\|T - \hat{T} - \Delta\hat{T}\|_1/\|T\|_1$. In Figure 4, we can see that for all cases, the proposed risk-consistent-estimator-based revision leads to smaller estimator errors than the classifier-consistent algorithm based method (Forward-R), showing that the risk-consistent estimator is more powerful in modifying the transition matrix. This also explains why the proposed method works better. We provide more discussions about Figure 4 in Appendix E.

## 5 Conclusion

This paper presents a risk-consistent estimator for label-noise learning without involving the inverse of transition matrix and a simple but effective learning paradigm called $T$-revision, which trains deep neural networks robustly under noisy supervision. The aim is to maintain effectiveness and efficiency of current consistent algorithms when there are no anchor points and then the transition matrices are poorly learned. The key idea is to revise the learned transition matrix and validate the revision by exploiting a noisy validation set. We conduct experiments on both synthetic and real-world label noise data to demonstrate that the proposed $T$-revision can significantly help boost the performance of label-noise learning. In the future, we will extend the work in the following aspects. First, how to incorporate some prior knowledge of the transition matrix, e.g., sparsity, into the end-to-end learning system. Second, how to recursively learn the transition matrix and classifier as our experiments show that transition matrices can be refined.

## Acknowledgments

TLL was supported by Australian Research Council Project DP180103424 and DE190101473. NNW was supported by National Natural Science Foundation of China under Grants 61922066, 61876142, and the CCF-Tencent Open Fund. CG was supported by NSF of China under Grants 61602246, 61973162, NSF of Jiangsu Province under Grants BK20171430, the Fundamental Research Funds for the Central Universities under Grants 30918011319, and the "Young Elite Scientists Sponsorship Program" by CAST under Grants 2018QNRC001. MS was supported by the International Research Center for Neurointelligence (WPI-IRCN) at The University of Tokyo Institutes for Advanced Study. XBX and TLL would give special thanks to Haifeng Liu and Brain-Inspired Technology Co., Ltd. for their support of GPUs used for this research.

## Footnotes

[1]In the literature, the assumption $\inf_x P(Y = i|X = x) \to 1$ was introduced as irreducibility [5] to ensure the transition matrix is identifiable; an anchor point $x$ for class $i$ is defined by $P(Y = i|X = x) = 1$ [35, 22] to ensure a fast convergence rate. In this paper, we generalize the definition for the anchor point family, including instances whose class posterior probability $P(Y = i|X = x)$ is equal to or close to one.

[2]During the training, $T + \Delta T$ can be ensured to be a valid transition matrix by first projecting their negative entries to be zero and then performing row normalization. In the experiments, $\Delta T$ is initialized to be a zero matrix and we haven't pushed $T + \Delta T$ to be a valid matrix when tuning $\Delta T$.

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
