[Supplementary Material]

# A  Label-noise learning, or noisy-label learning, that is the point

Note that the title of this paper is a question "are anchor points really indispensable in *label-noise learning*" but not "are anchor points really indispensable in *noisy-label learning*". Here, we explain why in order to make sense it must be label-noise learning. At first glance, label-noise learning may sound like we are learning the label noise, but this is exactly what we have implied in the title.

Generally speaking, the two names are synonyms of *learning with noisy labels*—this is the title of [26] where the first statistically consistent learning method was proposed for training classifiers with noisy labels. For the family of consistent learning methods, the estimation of the transition matrix $T$ is always necessary. In fact, any such method requires three components:

- a label corruption process parameterized by $T$,
- an estimator of $T$, and
- a statistical or algorithmic correction using the estimated $T$.

As a result, $T$ is also a target of learning, that is, the label noise is both a target to be learned and a source from which we learn. This is learning *with* noisy labels, more than learning *from* noisy labels.

For the first component, we assume the class-conditional noise model [26, 30]; for the third component, we rely on importance reweighting for learning with noisy labels [22]. Our novelty and major contribution is the second component, where we relax a requirement in existing consistent learning methods, namely we should have a certain amount of anchor points for estimating $T$ accurately, so that the correction using the estimated $T$ can be performed well.

While it is necessary to estimate $T$ for consistent learning methods, it is not the case for inconsistent learning methods. For instance, *sample selection* methods try to remove mislabeled data, and *label correction* methods try to fix the wrong labels of mislabeled data. None of them estimate $T$ so that none of them ever need the existence of anchor points.

Therefore, if we ask "are anchor points really indispensable in label-noise learning" where we are learning, modeling or estimating the label noise, the answer was *yes* previously and is *no* currently. Nevertheless, if we ask "are anchor points really indispensable in noisy-label learning" where some inconsistent learning method is employed without estimating the label noise, the answer has already been known to be *no*. That is the point.

# B  How consistent algorithms work

The aim of multi-class classification is to learn a *hypothesis* $f$ that predicts labels for given instances. Typically, the hypothesis is of the following form: $f(x) = \arg\max_{i \in \{1,2,...,C\}} g_i(x)$, where $g_i(x)$ is an estimate of $P(Y = i|X = x)$. Let define the *expected risk* of employing $f$ as

$$R(f) = \mathbb{E}_{(X,Y)\sim D}[\ell(f(X), Y)]. \tag{4}$$

The *optimal* hypothesis to learn is the one that minimizes the risk $R(f)$. Usually, the distribution $D$ is unknown. The optimal hypothesis is approximated by the minimizer of an *empirical counterpart* of $R(f)$, i.e., the empirical risk

$$R_n(f) = \frac{1}{n}\sum_{i=1}^{n} \ell(f(x_i), y_i). \tag{5}$$

The empirical risk $R_n(f)$ is **risk-consistent** w.r.t. all loss functions, i.e., $R_n(f) \rightarrow R(f)$ as $n \rightarrow \infty$. Note that in the main paper, we have treated the training sample $\{X_i, \bar{Y}_i\}_{i=1}^{n}$ as iid variables to derive the generalization bound.

If the loss function is zero-one loss, i.e., $\ell(f(x), y) = 1_{\{f(x)\neq y\}}$ where $1_{\{\cdot\}}$ is the indicator function and that the predefined *hypothesis class* [25] is large enough, the optimal hypothesis that minimizing $R(f)$ is identical to the Bayes classifier [3], i.e.,

$$f_\rho(x) = \arg\max_{i \in \{1,2,...,C\}} P(Y = i|X = x). \tag{6}$$

Many frequently used loss functions are proven to be *classification-calibrated* [3, 34], which means they will lead to classifiers having the same predictions as the classifier learned by using zero-one

loss if the training sample size is sufficiently large [42, 25]. In other words, the approximation, i.e., $\arg\min R_n(f)$, could converge to the optimal hypothesis by increasing the sample size $n$ and the corresponding estimator is therefore *classifier-consistent*. Note that risk-consistent algorithm is also classifier-consistent. However, a classifier-consistent algorithm may not be risk-consistent.

Given only the noisy training sample $\{(X_i, \bar{Y}_i)\}_{i=1}^n$, we have a noisy version of the empirical risk as

$$\bar{R}_n(f) = \frac{1}{n}\sum_{i=1}^n \ell(f(X_i), \bar{Y}_i). \tag{7}$$

The learned $g(X)$ can be used to approximate $P(\bar{\mathbf{Y}}|X)$. According to the definition of transition matrix, we have that $P(\bar{\mathbf{Y}}|X) = T^\top P(\mathbf{Y}|X)$, implying that if we let

$$\bar{h}(X) = \arg\max_{i\in\{1,2,\dots,C\}} (T^\top g)_i(X), \tag{8}$$

minimizing

$$\bar{R}_n(\bar{h}) = \frac{1}{n}\sum_{i=1}^n \ell(\bar{h}(X_i), \bar{Y}_i) \tag{9}$$

by using only noisy data will lead to a classifier-consistent algorithm. In other words, $\arg\max_{i\in\{1,2,\dots,C\}} g_i(x)$ in the algorithm will converge to the optimal classifier for clean data by increasing the noisy sample size. That's why noise adaption layer has been widely used in deep learning to modify the softmax function (i.e., $g(x)$) [9, 30, 38, 47].

If the transition matrix is invertible, the equation $P(\mathbf{Y}|X) = (T^\top)^{-1}P(\bar{\mathbf{Y}}|X)$ has been explored to design risk-consistent estimator for $R(f)$, e.g., [26, 30]. The basic idea is to modify the loss function $\ell(f(X), \bar{Y})$ to be $\tilde{\ell}(f(X), \bar{Y})$ such that for $X$ and $Y$,

$$\mathbb{E}_{\bar{Y}}[\tilde{\ell}(f(X), \bar{Y})] = \ell(f(X), Y) \tag{10}$$

and thus

$$\mathbb{E}_{(X,Y,\bar{Y})}\tilde{\ell}(f(X), \bar{Y}) = R(f). \tag{11}$$

Specifically, let

$$\mathcal{L}(f(X), \mathbf{Y}) = [\ell(f(X), Y=1),\dots,\ell(f(X), Y=C)]^\top \tag{12}$$

and

$$\tilde{\mathcal{L}}(f(X), \bar{\mathbf{Y}}) = [\tilde{\ell}(f(X), \bar{Y}=1),\dots,\tilde{\ell}(f(X), \bar{Y}=C)]^\top = (T^\top)^{-1}\mathcal{L}(f(X), \bar{\mathbf{Y}}). \tag{13}$$

The losses $\tilde{\ell}$ will lead to risk-consistent estimator because

$$\mathbb{E}_{\bar{Y}|Y}[\tilde{\mathcal{L}}(f(X), \bar{\mathbf{Y}})] = T^\top \tilde{\mathcal{L}}(f(X), \bar{\mathbf{Y}}) = \mathcal{L}(f(X), \mathbf{Y}). \tag{14}$$

Risk-consistent algorithms are also classifier-consistent, but have some unique properties than classifier-consistent algorithms, e.g., can be used to tune hyper-parameter. However, the current risk-consistent estimators contain the inverse of transition matrix, making parameter tuning inefficient and leading to performance degeneration. Our proposed risk-consistent estimator overcome the aforementioned issues.

## C   Proof of Theorem 1

We have defined

$$\bar{R}_{n,w}(\hat{T} + \Delta T, f) = \frac{1}{n}\sum_{i=1}^n \frac{g_{\bar{Y}_i}(X_i)}{((\hat{T} + \Delta T)^\top g)_{\bar{Y}_i}(X_i)}\ell(f(X_i), \bar{Y}_i), \tag{15}$$

where $f(X) = \arg\max_{i\in\{1,\dots,C\}} g_i(X)$. Let $S = \{(X_1, \bar{Y}_1),\dots,(X_n, \bar{Y}_n)\}$, $S^i = \{(X_1, \bar{Y}_1),\dots,(X_{i-1}, \bar{Y}_{i-1}),(X_i', \bar{Y}_i'),(X_{i+1}, \bar{Y}_{i+1}),\dots,(X_n, \bar{Y}_n)\}$, and

$$\Phi(S) = \sup_{\Delta T, f} (\bar{R}_{n,w}(\hat{T} + \Delta T, f) - \mathbb{E}_S[\bar{R}_{n,w}(\hat{T} + \Delta T, f)]). \tag{16}$$

**Lemma 1** *Let $\Delta\hat{T}$ and $\hat{f}$ be the learned slack variable and classifier respectively. Assume the learned transition matrix is valid, i.e., $\hat{T}_{ij} + \Delta\hat{T}_{ij} \geq 0$ for all $i, j$ and $\hat{T}_{ii} + \Delta\hat{T}_{ii} > \hat{T}_{ij} + \Delta\hat{T}_{ij}$ for all $j \neq i$. For any $\delta > 0$, with probability at least $1 - \delta$, we have*

$$\mathbb{E}[\bar{R}_{n,w}(\hat{T} + \Delta\hat{T}, \hat{f})] - \bar{R}_{n,w}(\hat{T} + \Delta\hat{T}, \hat{f}) \leq \mathbb{E}[\Phi(S)] + CM\sqrt{\frac{\log 1/\delta}{2n}}. \tag{17}$$

Detailed proof of Lemma 1 is provided in Section C.1.

Using the same trick to derive Rademacher complexity [4], we have

$$\mathbb{E}[\Phi(S)] \leq 2\mathbb{E}\left[\sup_{\Delta T, f} \frac{1}{n}\sum_{i=1}^{n} \sigma_i \frac{g_{\bar{Y}_i}(X_i)}{((\hat{T} + \Delta T)^\top g)_{\bar{Y}_i}(X_i)} \ell(f(X_i), \bar{Y}_i)\right], \tag{18}$$

where $\sigma_1, \ldots, \sigma_n$ are i.i.d. Rademacher random variables.

We can upper bound the right hand part of the above inequality by the following lemma.

**Lemma 2**

$$\mathbb{E}\left[\sup_{\Delta T, f} \frac{1}{n}\sum_{i=1}^{n} \sigma_i \frac{g_{\bar{Y}_i}(X_i)}{((\hat{T} + \Delta T)^\top g)_{\bar{Y}_i}(X_i)} \ell(f(X_i), \bar{Y}_i)\right] \leq \mathbb{E}\left[\sup_{f} \frac{1}{n}\sum_{i=1}^{n} \sigma_i \ell(f(X_i), \bar{Y}_i)\right]. \tag{19}$$

Note that Lemma 2 is not an application of Talagrand Contraction Lemma [20]. Detailed proof of Lemma 2 is provided in Section C.2.

Recall that $f = \arg\max_{i \in \{1,\ldots,C\}} g_i$ is the classifier, where $g$ is the output of the softmax function, i.e., $g_i(X) = \exp(h_i(X))/\sum_{k=1}^{C} \exp(h_k(X)), i = 1, \ldots, C$, and $h(X)$ is defined by a $d$-layer neural network, i.e., $h : X \mapsto W_d \sigma_{d-1}(W_{d-1}\sigma_{d-2}(\ldots\sigma_1(W_1 X))) \in \mathbb{R}^C$, $W_1, \ldots, W_d$ are the parameter matrices, and $\sigma_1, \ldots, \sigma_{d-1}$ are activation functions. To further upper bound the Rademacher complexity, we need to consider the Lipschitz continuous property of the loss function w.r.t. to $h(X)$. To avoid more assumption, We discuss the widely used *cross-entropy loss*, i.e.,

$$\ell(f(X), \bar{Y}) = -\sum_{i=1}^{C} 1_{\{\bar{Y}=i\}} \log(g_i(X)). \tag{20}$$

We can further upper bound the Rademacher complexity by the following lemma.

**Lemma 3**

$$\mathbb{E}\left[\sup_{f} \frac{1}{n}\sum_{i=1}^{n} \sigma_i \ell(f(X_i), \bar{Y}_i)\right] \leq CL\mathbb{E}\left[\sup_{h \in H} \frac{1}{n}\sum_{i=1}^{n} \sigma_i h(X_i)\right], \tag{21}$$

*where $H$ is the function class induced by the deep neural network.*

Detailed proof of Lemma 3 is provided in Section C.3.

Note that $\mathbb{E}\left[\sup_{h \in H} \frac{1}{n}\sum_{i=1}^{n} \sigma_i h(X_i)\right]$ measures the hypothesis complexity of deep neural networks, which has been widely studied recently [27, 2, 10, 28]. Specifically, [10] proved the following theorem (Theorem 1 therein).

**Theorem 2** *Assume the Frobenius norm of the weight matrices $W_1, \ldots, W_d$ are at most $M_1, \ldots, M_d$. Let the activation functions be 1-Lipschitz, positive-homogeneous, and applied element-wise (such as the ReLU). Let $x$ is upper bounded by $B$, i.e., for any $x \in \mathcal{X}, \|x\| \leq B$. Then,*

$$\mathbb{E}\left[\sup_{h \in H} \frac{1}{n}\sum_{i=1}^{n} \sigma_i h(X_i)\right] \leq \frac{B(\sqrt{2d\log 2} + 1)\Pi_{i=1}^{d} M_i}{\sqrt{n}}. \tag{22}$$

Theorem 1 follows by combining Lemmas 1, 2, 3, and Theorem 2.

## C.1 Proof of Lemma 1

We employ McDiarmid's concentration inequality [7] to prove the lemma. We first check the bounded difference property of $\Phi(S)$, e.g.,

$$\Phi(S) - \Phi(S^i) \leq \sup_{\Delta T, f} \frac{1}{n} \left( \frac{g_{\bar{Y}_i}(X_i)\ell(f(X_i), \bar{Y}_i)}{((\hat{T} + \Delta T)^\top g)_{\bar{Y}_i}(X_i)} - \frac{g_{\bar{Y}'_i}(X'_i)\ell(f(X'_i), \bar{Y}'_i)}{((\hat{T} + \Delta T)^\top g)_{\bar{Y}'_i}(X'_i)} \right). \tag{23}$$

Before further upper bounding the above difference, we show that the weighted loss is upper bounded by $CM$. Specifically, we have assume the learned transition matrix is valid, i.e., $\hat{T}_{ij} + \Delta T_{ij} \geq 0$ for all $i, j$ and $\hat{T}_{ii} + \Delta T_{ii} > \hat{T}_{ij} + \Delta T_{ij}$ for all $j \neq i$. Thus $\frac{g_{\bar{Y}}(X)}{((\hat{T}+\Delta T)^\top g)_{\bar{Y}}(X)} \leq 1/\min_i(\hat{T}_{ii} + \Delta T_{ii}) \leq C$ for any $(X, \bar{Y})$ and $\hat{g}$. Then, we can conclude that the weighted loss is upper bounded by $CM$ and that

$$\Phi(S) - \Phi(S^i) \leq \frac{CM}{n}. \tag{24}$$

Similarly, we could prove that $\Phi(S^i) - \Phi(S) \leq \frac{CM}{n}$.

By employing McDiarmid's concentration inequality, for any $\delta > 0$, with probability at least $1 - \delta$, we have

$$\Phi(S) - \mathbb{E}[\Phi(S)] \leq CM\sqrt{\frac{\log(1/\delta)}{2n}}. \tag{25}$$

## C.2 Proof of Lemma 2

Given the learned transition matrix is valid, we have shown that $\frac{g_{\bar{Y}}(X)}{((\hat{T}+\Delta T)^\top g)_{\bar{Y}}(X)} \leq 1/\min_i(\hat{T}_{ii} + \Delta T_{ii}) \leq C$ for all $(X, \bar{Y})$ in the proof of Lemma 1.

Lemma 2 holds of we could prove the following inequality

$$\mathbb{E}_\sigma \left[ \sup_{\Delta T, f} \frac{1}{n} \sum_{i=1}^n \sigma_i \frac{g_{\bar{Y}_i}(X_i)}{((\hat{T} + \Delta T)^\top g)_{\bar{Y}_i}(X_i)} \ell(f(X_i), \bar{Y}_i) \right] \leq \mathbb{E}_\sigma \left[ \sup_f \frac{1}{n} \sum_{i=1}^n \sigma_i \ell(f(X_i), \bar{Y}_i) \right]. \tag{26}$$

Note that

$$\begin{aligned}
&\mathbb{E}_\sigma \left[ \sup_{\Delta T, f} \frac{1}{n} \sum_{i=1}^n \sigma_i \frac{g_{\bar{Y}_i}(X_i)}{((\hat{T} + \Delta T)^\top g)_{\bar{Y}_i}(X_i)} \ell(f(X_i), \bar{Y}_i) \right] \\
&= \mathbb{E}_{\sigma_1, \dots, \sigma_{n-1}} \left[ \mathbb{E}_{\sigma_n} \left[ \sup_{\Delta T, f} \frac{1}{n} \sum_{i=1}^n \sigma_i \frac{g_{\bar{Y}_i}(X_i)}{((\hat{T} + \Delta T)^\top g)_{\bar{Y}_i}(X_i)} \ell(f(X_i), \bar{Y}_i) \right] \right].
\end{aligned} \tag{27}$$

Let $s_{n-1}(\Delta T, f) = \sum_{i=1}^{n-1} \sigma_i \frac{g_{\bar{Y}_i}(X_i)}{((\hat{T}+\Delta T)^\top g)_{\bar{Y}_i}(X_i)} \ell(f(X_i), \bar{Y}_i)$.

By definition of the supremum, for any $\epsilon > 0$, there exist $(\Delta T, f_1)$ and $(\Delta T, f_2)$ such that

$$\begin{aligned}
&\frac{g_{\bar{Y}_n}(X_n)}{((\hat{T} + \Delta T)^\top g)_{\bar{Y}_n}(X_n)} \ell(f_1(X_n), \bar{Y}_n) + s_{n-1}(\Delta T, f_1) \\
&\geq (1 - \epsilon) \sup_{\Delta T, f} \left( \frac{g_{\bar{Y}_n}(X_n)}{((\hat{T} + \Delta T)^\top g)_{\bar{Y}_n}(X_n)} \ell(f(X_n), \bar{Y}_n) + s_{n-1}(\Delta T, f) \right)
\end{aligned} \tag{28}$$

and

$$\begin{aligned}
&- \frac{g_{\bar{Y}_n}(X_n)}{((\hat{T} + \Delta T)^\top g)_{\bar{Y}_n}(X_n)} \ell(f_2(X_n), \bar{Y}_n) + s_{n-1}(\Delta T, f_2) \\
&\geq (1 - \epsilon) \sup_{\Delta T, f} \left( - \frac{g_{\bar{Y}_n}(X_n)}{((\hat{T} + \Delta T)^\top g)_{\bar{Y}_n}(X_n)} \ell(f(X_n), \bar{Y}_n) + s_{n-1}(\Delta T, f) \right).
\end{aligned} \tag{29}$$

Thus, for any $\epsilon$, we have

$$(1-\epsilon)\mathbb{E}_{\sigma_n}\left[\sup_{\Delta T, f}\left(\sigma_n\frac{g_{\bar{Y}_n}(X_n)}{((\hat{T}+\Delta T)^\top g)_{\bar{Y}_n}(X_n)}\ell(f(X_n),\bar{Y}_n)+s_{n-1}(\Delta T,f)\right)\right]$$

$$=\frac{(1-\epsilon)}{2}\sup_{\Delta T,f}\left(\frac{g_{\bar{Y}_n}(X_n)}{((\hat{T}+\Delta T)^\top g)_{\bar{Y}_n}(X_n)}\ell(f_1(X_n),\bar{Y}_n)+s_{n-1}(\Delta T,f_1)\right)$$

$$+\frac{(1-\epsilon)}{2}\sup_{\Delta T,f}\left(-\frac{g_{\bar{Y}_n}(X_n)}{((\hat{T}+\Delta T)^\top g)_{\bar{Y}_n}(X_n)}\ell(f_2(X_n),\bar{Y}_n)+s_{n-1}(\Delta T,f_2)\right) \tag{30}$$

$$\leq\frac{1}{2}\left(\frac{g_{\bar{Y}_n}(X_n)}{((\hat{T}+\Delta T)^\top g)_{\bar{Y}_n}(X_n)}\ell(f_1(X_n),\bar{Y}_n)+s_{n-1}(\Delta T,f_1)\right.$$

$$\left.+(s_{n-1}(\Delta T,f_2)-\frac{g_{\bar{Y}_n}(X_n)}{((\hat{T}+\Delta T)^\top g)_{\bar{Y}_n}(X_n)}\ell(f_2(X_n),\bar{Y}_n))\right)$$

$$\leq\frac{1}{2}\left(s_{n-1}(\Delta T,f_1)+s_{n-1}(\Delta T,f_2)+C|\ell(f_1(X_n),\bar{Y}_n)-\ell(f_2(X_n),\bar{Y}_n)|\right),$$

where the last inequality holds because $\frac{g_{\bar{Y}}(X)}{((\hat{T}+\Delta T)^\top g)_{\bar{Y}}(X)}\leq C$ for any $(X,\bar{Y}),g$, and valid $\hat{T}+\Delta T$.

Let $s=\mathrm{sgn}(\ell(f_1(X_n),\bar{Y}_n)-\ell(f_2(X_n),\bar{Y}_n))$. We have

$$(1-\epsilon)\mathbb{E}_{\sigma_n}\left[\sup_{\Delta T,f}\left(\sigma_n\frac{g_{\bar{Y}_n}(X_n)}{((\hat{T}+\Delta T)^\top g)_{\bar{Y}_n}(X_n)}\ell(f(X_n),\bar{Y}_n)+s_{n-1}(\Delta T,f)\right)\right]$$

$$\leq\frac{1}{2}\left(s_{n-1}(\Delta T,f_1)+s_{n-1}(\Delta T,f_2)+sC(\ell(f_1(X_n),\bar{Y}_n)-\ell(f_2(X_n),\bar{Y}_n))\right)$$

$$=\frac{1}{2}\left(s_{n-1}(\Delta T,f_1)+sC\ell(f_1(X_n),\bar{Y}_n)\right)+\frac{1}{2}\left(s_{n-1}(\Delta T,f_2)-sC\ell(f_2(X_n),\bar{Y}_n)\right) \tag{31}$$

$$\leq\frac{1}{2}\sup_{f\in F}\left(s_{n-1}(\Delta T,f)+sC\ell(f(X_n),\bar{Y}_n)\right)+\frac{1}{2}\sup_{f\in F}\left(s_{n-1}(\Delta T,f)-sC\ell(f(X_n),\bar{Y}_n)\right)$$

$$=\mathbb{E}_{\sigma_n}\left[\sup_{\Delta T,f}\left(\sigma_n\ell(f(X_n),\bar{Y}_n)+s_{n-1}(\Delta T,f)\right)\right].$$

Since the above inequality holds for any $\epsilon>0$, we have

$$\mathbb{E}_{\sigma_n}\left[\sup_{\Delta T,f}\left(\sigma_n\frac{g_{\bar{Y}_n}(X_n)}{((\hat{T}+\Delta T)^\top g)_{\bar{Y}_n}(X_n)}\ell(f(X_n),\bar{Y}_n)+s_{n-1}(\Delta T,f)\right)\right]$$

$$\leq\mathbb{E}_{\sigma_n}\left[\sup_{\Delta T,f}\left(\sigma_n\ell(f(X_n),\bar{Y}_n)+s_{n-1}(\Delta T,f)\right)\right]. \tag{32}$$

Proceeding in the same way for all other $\sigma$, we have

$$\mathbb{E}_{\sigma}\left[\sup_{\Delta T,f}\sum_{i=1}^n\sigma_i\frac{g_{\bar{Y}_i}(X_i)}{((\hat{T}+\Delta T)^\top g)_{\bar{Y}_i}(X_i)}\ell(f(X_i),\bar{Y}_i)\right]\leq\mathbb{E}_{\sigma}\left[\sup_{f\in F}\sum_{i=1}^n\sigma_i\ell(f(X_i),\bar{Y}_i)\right]. \tag{33}$$

and thus

$$\mathbb{E}\left[\sup_{\Delta T,f}\sum_{i=1}^n\sigma_i\frac{g_{\bar{Y}_i}(X_i)}{((\hat{T}+\Delta T)^\top g)_{\bar{Y}_i}(X_i)}\ell(f(X_i),\bar{Y}_i)\right]\leq\mathbb{E}\left[\sup_{f\in F}\sum_{i=1}^n\sigma_i\ell(f(X_i),\bar{Y}_i)\right]. \tag{34}$$

### C.3 Proof of Lemma 3

Before proving Lemma 3, we show that the loss function $\ell(f(X),\bar{Y})$ is 1-Lipschitz-continuous w.r.t. $h_i(X), i=\{1,\ldots,C\}$.

Recall that

$$\ell(f(X), \bar{Y}) = -\sum_{i=1}^{C} 1_{\{\bar{Y}=i\}} \log(g_i(X)) = -\log\left(\frac{\exp(h_{\bar{Y}}(X))}{\sum_{i=1}^{C} \exp(h_i(X))}\right). \tag{35}$$

Take the derivative of $\ell(f(X), \bar{Y})$ w.r.t. $h_i(X)$. If $i \neq \bar{Y}$, we have

$$\frac{\partial \ell(f(X), \bar{Y})}{\partial h_i(X)} = \frac{\exp(h_i(X))}{\sum_{i=1}^{c} \exp(h_i(X))}. \tag{36}$$

If $i = \bar{Y}$, we have

$$\frac{\partial \ell(f(X), \bar{Y})}{\partial h_i(X)} = -1 + \frac{\exp(h_i(X))}{\sum_{i=1}^{c} \exp(h_i(X))}. \tag{37}$$

According to Eqs.(36) and (37), it is easy to conclude that $-1 \leq \frac{\partial \ell(f(X), \bar{Y})}{\partial h_i(X)} \leq 1$, which also indicates that the loss function is 1-Lipschitz with respect to $h_i(X), \forall i \in \{1, \ldots, C\}$.

Now we are ready to prove Lemma 3. We have

$$
\begin{aligned}
&\mathbb{E}\left[\sup_{f} \frac{1}{n}\sum_{i=1}^{n} \sigma_i \ell(f(X_i), \bar{Y}_i)\right] \\
&= \mathbb{E}\left[\sup_{f=\arg\max\{h_1,\ldots,h_c\}} \frac{1}{n}\sum_{i=1}^{n} \sigma_i \ell(f(X_i), \bar{Y}_i)\right] \\
&= \mathbb{E}\left[\sup_{\max\{h_1,\ldots,h_c\}} \frac{1}{n}\sum_{i=1}^{n} \sigma_i \ell(f(X_i), \bar{Y}_i)\right] \\
&\leq \mathbb{E}\left[\sum_{k=1}^{C} \sup_{h_k \in H} \frac{1}{n}\sum_{i=1}^{n} \sigma_i \ell(f(X_i), \bar{Y}_i)\right] \\
&= \sum_{k=1}^{C} \mathbb{E}\left[\sup_{h_k \in H} \frac{1}{n}\sum_{i=1}^{n} \sigma_i \ell(f(X_i), \bar{Y}_i)\right] \\
&\leq CL\mathbb{E}\left[\sup_{h_k \in H} \frac{1}{n}\sum_{i=1}^{n} \sigma_i h_k(X_i)\right] \\
&= CL\mathbb{E}\left[\sup_{h \in H} \frac{1}{n}\sum_{i=1}^{n} \sigma_i h(X_i)\right],
\end{aligned}
\tag{38}
$$

where the first equation holds because the softmax function preserves the rank of its inputs, i.e., $f(X) = \arg\max_{i \in \{1,\ldots,C\}} g_i(X) = \arg\max_{i \in \{1,\ldots,C\}} h_i(X)$; the second equation holds because $\arg\max\{h_1, \cdots, h_c\}$ and $\max\{h_1, \cdots, h_c\}$ give the same constraint on $h_i, \forall i \in \{1, \ldots, C\}$; the fifth inequality holds because of the Talagrand Contraction Lemma [20].

## D   Definition of transition matrix

The definition of symmetry flipping transition matrix is as follows, where $C$ is number of the class.

$$
\text{sym-}\epsilon: \quad T = \begin{bmatrix}
1-\epsilon & \frac{\epsilon}{C-1} & \cdots & \frac{\epsilon}{C-1} & \frac{\epsilon}{C-1} \\
\frac{\epsilon}{C-1} & 1-\epsilon & \frac{\epsilon}{C-1} & \cdots & \frac{\epsilon}{C-1} \\
\vdots & & \ddots & & \vdots \\
\frac{\epsilon}{C-1} & \cdots & \frac{\epsilon}{C-1} & 1-\epsilon & \frac{\epsilon}{C-1} \\
\frac{\epsilon}{C-1} & \frac{\epsilon}{C-1} & \cdots & \frac{\epsilon}{C-1} & 1-\epsilon
\end{bmatrix}.
$$

Figure 4: Comparing the estimation error of the transition matrix by employing classifier-consistent and risk-consistent estimators. The first row is about sym-20 label noise while the second row is about sym-50 label noise. The error bar for STD in each figure has been highlighted as a shade.

# E More discussions about Figure 3

We represent Figure 3 in the main paper as Figure 1 in this appendix.

From the figure, we can compare the transition matrices learned by the proposed T-revision method and the traditional anchor point based method. Specifically, as shown in Figure 1, at epoch 0, the estimation error corresponds to the estimation error of transition matrix learned by identifying anchor points [38] (the traditional method to learn transition matrix). Note that the method with "-N/A" in its name means it runs on the modified datasets where instances with large clean class posterior probobilities are removed (anchor points are removed); while the method with "-A" in its name means it runs the original intact dataset (may contain anchor points). Clearly, we can see that the estimation error will increase by removing possible anchor points, meaning that anchor points is crucial in the traditional transition matrix learning. Moreover, as the number of epochs grows, the figures show how the estimation error varies by running the proposed revision methods. We can see that the proposed Reweight method always leads to smaller estimation errors, showing that the proposed method works well in find a better transition matrix.

Figure 1 also shows the comparison of learning transition matrices between the risk-consistent estimator based method and the classifier-consistent method based method. For classifier-consistent algorithms, we can also modify the transition matrix by adding a slack variable and learning it jointly with the classifier, e.g., Forward-A-R and Forward-N/A-R. However, we can find that the classifier-consistent algorithm based method Forward-N/A-R may fail in learning a good transition matrix, e.g., Figure 1(a). This is because there is no reason to learn the transition matrix by minimizing the classifier-consistent objective function. It is reasonable to learn the transition matrix by minimizing the risk-consistent estimator because a favorable transition matrix should make the classification risk w.r.t. clean data small. This is verified by comparing Forward-A-R and Forward-N/A-R with the proposed Reweight-A-R and Reweight-N/A-R, we can find that the risk-consistent estimator Reweight always leads to smaller estimation errors for learning transition matrix.