[Reviews · NeurIPS 2019]

Reviewer 1



The paper proposes a new method for learning a label noise transition matrix from a noisy dataset. Unlike previously proposed approaches (e.g. the forward method) the proposed approach is shown to be less dependent on anchor points, which are defined to be examples that would have a have the output of the softmax extremely focused on the correct class in the noiseless domain. The main idea is to use importance reweighting to account for the discrepancy between the output of a noise free classifier and a noisy one, and to allow learnable modifications to the initially estimated transition matrix. The experiments show modest improvements over the SoA on MNIST and CIFAR-10 under low and medium levels of label noise. When anchor points are purposefully removed, the improvements on CIFAR-10 and 100 are more substantial, illustrating that the approach is less dependent on the presence of anchor points. Originality: The improvement over the forward method is somewhat incremental, but the idea is novel as far as I know. Clarity: There are a few language errors in the paper, but overall it is well written and the explanation of the approach is clear. Significance: While the results on CIFAR-10 look good, the results on CIFAR-100 are a little less convincing, especially in the case of 50% label noise, especially given the overlapping error bars. The results on Clothing1M demonstrate a clear improvement on the SoA.

Reviewer 2



The paper adresses an aimportant problem and proposes an original approach allowing to avoid the use of anchor point The method is both assess theoritically and experimentally.

Reviewer 3



This paper studies the challenging problem of learning from data with noisy labels. In particular, a novel transition revision (T-Revision) method is proposed, which does not require anchor points. T-Revision can effectively learn transition matrices which lead to better classifiers. A deep-learning-based risk-consistent estimator is designed to tune transition matrix accurately. Experimental results on multiple benchmark datasets show that T-Revision outperforms the state-of-the-art methods. This paper makes a significant contribution to the label-noise learning problem. The proposed method is well motivated and clearly presented. Technical details are easy to follow, and theoretical analysis on generalization error is provided. Moreover, the implementation details of the proposed method are also provided, which will be very helpful in reproducing the reported results. My comments are as follows. 1. The difference between two categories of label-noise learning algorithms has been mentioned in Section 1. It will be helpful if the authors can elaborate more about the advantages of risk-/classifier-consistent algorithms. 2. In the experiments, the Sym-50 setting on MNIST can still obtain a good performance (about 98%). I wonder what the performance would be in extreme cases with even more label noise. 3. I'm curious if the proposed mechanism can be easily extended to semi-supervised setting.

[Author Response · NeurIPS 2019]

Dear all reviewers: Thank you very much for taking your time to review our paper and providing us valuable and insightful comments. Below, we answer all of the questions.

**R1** Q1) It wasn't totally clear to me how you ensure $T + \Delta T$ remains a valid transition matrix throughtout training. Also, is $\Delta T$ initialized to zero matrix, or randomly?

A1) We ensure $T + \Delta T$ to be a valid transition matrix by first projecting their negative entries to be zero and then performing row normalization. About $\Delta T$, we initialize it to be a zero matrix in our experiments.

**R1** Q2) Can this approach take advantage of a small clean set?

A2) Yes. If a small clean set is available, it is helpful. It can be used to (1) better initialize the transition matrix, (2) better validate the slack variable $\Delta T$, and to (3) fine-tune the deep network.

**R1** Q3) **It would be interesting to also include results using a subset of the WebVision dataset to see if the method works there too.**

A3) **Due to the limited time, we do experiments on a subset of WebVision dataset**. Specifically, we create the training data by sampling a hundred classes from the thousand classes and sampling 1,600 images for each class. 10% of the training data is held out for validation. We use the total of 5,000 images from the sampled 100 classes in the original validation set as the test set. We compare "Forward", "Reweight", "Forward-R", "Reweight-R" on this subset. We use a ResNet-50 model pretrained on Imagenet. The results are 80.48% (Forward), 81.08% (Reweight), 85.12% (Forward-R), 85.42% (Reweight-R). We can see that "Reweight" and "Reweight-R" achieve better results and the revision technique greatly boost the performance. Due to the limited time, we only compared with "Foward" and "Reweighting" (in our setting, those two methods consistently work better than other baselines).

**R1** Q4) Discuss the relationship with "Multiclass learning with partially corrupted labels"(Wang et al)

A4) Thank you for the valuable feedback. They both employ the importance reweighting technique. However, their approach requires anchor points to estimate the transition matrix; while the proposed approach is designed to release the strong requirement of anchor points.

**R2** Q5) More experiments on real data.

A5) We perform our experiments on the subset (metioned in A3) of WebVision using four models. The results of experiments are 80.48% (Forward), 81.08% (Reweight), 85.12% (Forward-R), 85.42% (Reweight-R). More details can be found in A3.

**R3** Q6) Please discuss the model performance on MNIST with more label noise.

A6) We raise the noise rates to be 60%, 70%, 80%. Other experiment settings are unchanged. The results are presented in Tables 1 and 2. We can see that the proposed model outperforms the baselines more significantly as the noise rate grows. Due to the limited time, we only compared with "Foward" and "Reweighting" (in our setting, those two methods consistently work better than other baselines).

**R3** Q7) Discuss the potential extension of the proposed approach.

A7) We can extend our approach to mixture proportion estimation [27] and learning with complementary label.

Table 1: Mean test accuracy (in %, $\pm$ std dev).

|  | Sym-60% | Sym-70% | Sym-80% |
|---|---|---|---|
| Forward-A | 97.10±0.08 | 96.06±0.41 | 91.46±1.03 |
| Forward-A-R | 97.65±0.11 | 96.42±0.35 | 91.77±0.22 |
| Reweight-A | 97.39±0.27 | 96.25±0.26 | 93.79±0.52 |
| Reweight-A-R | **97.83±0.18** | **97.13±0.08** | **94.19±0.45** |

Table 2: Mean test accuracy (in %, $\pm$ std dev), anchor points removed.

|  | Sym-60% | Sym-70% | Sym-80% |
|---|---|---|---|
| Forward-N/A | 96.82±0.14 | 94.61±0.28 | 85.95±1.01 |
| Forward-N/A-R | 96.99±0.16 | 95.02±0.17 | 86.04±1.03 |
| Reweight-N/A | 97.01±0.20 | 95.94±0.14 | 91.59±0.70 |
| Reweight-N/A-R | **97.81±0.12** | **96.59±0.15** | **91.91±0.65** |

- "-A" means the transition matrix is estimated by using the instance $X$ with highest estimated $P(\bar{Y}|X)$ (which are likely to be anchor points).

- "-N/A" means instances with high estimated $P(Y|X)$ are removed from the dataset.

- "-R" means that the transition matrix used is revised by a revision $\Delta T$.

- The highest accuracy in each column is bold faced.

[Meta-Review · NeurIPS 2019]

Are Anchor Points Really Indispensable in Label-Noise Learning. The author provided a new algorithm to do this based on the idea of anchor points without requiring them. The empirical performance of the method was well-demonstrated and the reviewers are in consensus that this paper should be accepted.